# Design of Multi-Layer Graphene Membrane with Descending Pore Size for 100% Water Desalination by Simulation Using ReaxFF

**DOI:** 10.3390/membranes12111038

**Published:** 2022-10-25

**Authors:** Qusai Ibrahim, Rokhsareh Akbarzadeh, Salem S. Gharbia, Patrick Gathura Ndungu

**Affiliations:** 1Department of Mechanical Engineering Science, University of Johannesburg, Auckland Park Kingsway Campus, Johannesburg 2006, South Africa; 2Energy, Sensors and Multifunctional Nanomaterials Research Group, Department of Chemical Sciences, Faculty of Science, University of Johannesburg, Doornfontein 2028, South Africa; 3Department of Physics, Faculty of Science, University of South Bohemia, 370 05 České Budějovice, Czech Republic; 4Centre for Environmental Research Innovation and Sustainability CERIS, Department of Environmental Science, Atlantic Technological University, F91 YW50 Sligo, Ireland; 5Department of Chemistry, University of Pretoria, Pretoria 0028, South Africa

**Keywords:** graphene, membrane, ReaxFF, material studio, desalination, MD simulation

## Abstract

The performance of a desalination membrane depends on a specific pore size suitable for both water permeability and salt rejection. To increase membrane permeability, the applied pressure should be increased, which creates the need to improve membrane stability. In this research article, a molecular dynamics (MD) simulation was performed using ReaxFF module from Amsterdam Modeling suite (AMS) software to simulate water desalination efficiency using a single and multi-layer graphene membrane. The graphene membrane with different pore sizes and a multi-layer graphene membrane with descending pore size in each layer were designed and studied under different pressures. The stability of the membrane was checked using Material Studio 2019 by studying the dynamics summary. The single-layer graphene membrane was evaluated under pressures ranging from 100 to 500 MPa, with the salt rejection ranging from 95% to 82% with a water permeability of 0.347 × 10^−9^ to 2.94 × 10^−9^ (mm.g.cm^−2^s^−1^.bar^−1^), respectively. Almost 100% salt rejection was achieved for the multi-layer graphene membrane. This study successfully demonstrated the design and optimization of graphene membrane performance without functionalization.

## 1. Introduction

The increasing need for water and the shortage of high-quality drinking water have resulted in investigating methods to improve water purification technologies. With the abundance of sea and ocean water, desalination is one of the most promising methods to supply freshwater, which is defined as a process that separates the salt from saline water [1].

So far, reverse osmosis (RO) has been mainly used for water desalination. In the RO process, an external positive hydrostatic pressure applies to push the water through a semi-permeable membrane, which allows a larger volume of water to pass through the membrane while blocking dissolved salts and other impurities [2]. RO is considered one of the most energy-efficient technologies for desalination of water and has become a scale for evaluating any new desalination technology [3]. However, the level of energy consumption for RO has been reduced in the past few decades by improvements in membrane technology. The new membrane technologies offer higher permeability. However, RO still suffers from low desalination capacity and high capital costs [1]. In the desalination process, the role of the membrane is critical. Therefore, the development of more permeable membranes in RO can reduce energy consumption and consequently reduce running costs.

It has been proven that the designing of nanoporous graphene membranes can allow faster water flow across well-defined channels [4]. Nanoporous membranes with small dimensions can be used based on molecular size. In addition, there are different methods, such as electron beam, chemical etching and ion beam drilling, that can be used to create these pores [5]. Several theoretical and computational studies have been performed to predict nanoporous materials’ desalination performance; these materials have higher capacity for water desalination compared to available technologies [6]. During the past decade, graphene has attracted interest in different areas, including desalination [7]. For the first time, in 2013, Lockheed Martin Corporation patented a graphene-based membrane with a nano hole size for the desalination of water [8].

Although various materials such as polymers and ceramics have been used or proposed for membrane fabrication [9], graphene-based materials have recently emerged as potential candidates, with excellent desalination characteristics [5]. Graphene is an effective membrane because of its specific chemical and mechanical properties. In addition to the graphene permeability and salt rejection properties, graphene (G) membranes are less expensive than other membranes, such as MoS_2_, that have been used for desalination due to their higher salt rejection and water permeability [10,11].

Controlling the synthesis of graphene, maintaining high structural stability to withstand loading pressures, and the demand for accurate computational characterization are the challenges associated with the development of graphene-based membranes. Different methods and possibilities have been discussed and suggested by various researchers to push graphene and other 2D material membranes research works toward practical separation applications [7]. A detailed simulation of this kind of material can therefore open up a way to predict how to design the membrane and predict membrane behavior in real desalination applications.

As one of the recent materials under study for membrane development, graphene is a promising material for water desalination because of its two key advantages of permeability and selectivity [12]. A single-layer graphene membrane can effectively remove sodium chloride salt from water. However, various factors affect membrane efficiency, including design, pore size and applied pressure. The graphene-based membrane can be designed with different pore sizes to block NaCl salt while allowing water molecules to pass. Salt rejection increases when the pore size decreases, but this reduces the water permeability.

Different procedures and methods have been used to improve salt rejection and permeability. For instance, Nguyen and Beskok (2019) [13] found that charging of single-layer nanoporous graphene membranes, positively and negatively, could increase the salt rejection even with a high pore diameter. Hossieni et al. (2019) [14] improved the salt rejection of graphene by fluorine functionalization. Fischbein and Drndić [15] used an electron beam using a transmission electron microscope to create nanopores. They demonstrated that during the creation of nanopores, the size of the nanopore can be controlled with electron-beam irradiation, which offers a route to fabricate graphitic structures for potential use in electrical, mechanical and molecular translocation studies [15,16,17]. Another study by Lu et al. [18] demonstrated that a combination of electron beam irradiation and controlled heat could tailor the size of graphene pores. In 2018, Rikhtehgaran and his colleagues designed a multi-layer graphene membrane with different pore sizes for water desalination. They achieved 86% ion rejection with the pore radius of 3.3 Å and layer separation of 20 Å [19]. In our previous research work, we studied a composite of TiO_2_ and graphene, which showed an increase in membrane stability and a salt rejection of 98% under an applied pressure of 100 MPa [20].

Using molecular dynamic simulation, this study designed and evaluated a single and multi-layer graphene membrane with a descending pore size (from a bigger pore size in the first layer to a smaller pore size towards the bottom) for water permeability and salt rejection without functionalization or any change in the surface chemistry of the membrane. The performance of the designed graphene membrane for water desalination under certain water flux and pressure for different pore sizes was determined. The durability of the membrane was also evaluated.

## 2. Materials and Methods

### 2.1. Molecular Dynamics Simulation

According to Duran et al., [21] computational models are mathematical models used to numerically study complex systems’ behaviour through a computer simulation. This can help predict the system’s behaviour under different conditions, mainly for cases where analytical solutions are not available or not economically viable to repeat or conduct such an experiment. In this study, reactive force-field (ReaxFF) was used for molecular dynamics (MD) simulation. ReaxFF module of Amsterdam Modeling suite (AMS) which is a simulation tool developed and introduced by Duin and his colleagues for MD simulation [22].

In this study, single-layer and multi-layer graphene membranes were designed and investigated for water desalination. A simulation box consisting of a graphene sheet was designed as a membrane with 3300 fixed number of molecules (3000 H_2_O, and 300 NaCl) to apply an external force to the solution. NaCl solution with a concentration of 10% was chosen as the feed solution to model the saline water. A graphene membrane having a hexagonal honeycomb structure was designed and created using crystallographic parameters (a = 4.26, b = 2.46 and c = 25 Å), obtained from ReaxFF online library. The simulation box was designed with dimensions of 36 × 34 × 60 Å^3^ for a single-layer graphene membrane, with an angle of 60° between the lateral axis. For the multi-layer membrane, three graphene sheets with different pore sizes (2.841 Å, 4.261 Å and 7.101 Å) were placed in parallel, with a distance equal to 3.664 Å and with the same simulation box size. The NPT Berendsen simulation method [23] was used, which is efficient in equilibrating the MD simulation system. The force field values for graphene and the elements in NaCl and water elements were selected from the software library [24]. The temperature inside the simulation box was selected as 323.15 K, with a damping constant equal to 500 fs. The water permeability and salt rejection were studied for all pores sizes under the applied pressure of 100, 200, 300 and 500 MPa.

### 2.2. Investigations of the Properties of the Membrane with Recommended Pores

Figure 1 shows the dimensions of H_2_O and NaCl molecules. The radius of oxygen atoms is 1.517 Å, and the length of the hydrogen bond is 0.974 Å. For the NaCl molecules, the radii of Na and Cl atoms are 2.250 Å and 1.725 Å, respectively, in the molecule form. Several pore sizes were selected to study the membrane permeability and salt rejection capacity based on these values. In this study, graphene membranes with three different pore sizes were considered and simulated using ReaxFF. Initially, a graphene sheet with no pores was constructed, which considered a membrane with the original pore size of 2.841 Å and then the membrane with pore size of 4.261 and 7.101 Å were prepared. Figure 2 shows the diameter of pore size used in this study.

Figure 3 shows the atomic structure of the G membrane sheets with pore sizes of 2.841 (without creating pore), 4.261 and 7.101 Å. Then, the multi-layer graphene membrane was also designed and studied, including three graphene layers, with each layer having a different pore size.

By using Material studio 2019, the mechanical behaviour of graphene membranes with 2.8, 4.3, and 7.1 Å pore sizes was investigated under a 500 MPa applied pressure to check the membrane stability before and after creating the pores. A dynamic task was used by CASTEP code with the NPT Berendsen control method [23] for the duration of 500 ps (decay constant of 0.1 ps and 1 fs time steps). Figure 4 and Table 1 present the energy changes for the single and multi-layer graphene membranes.

As can be seen, the potential energy increased for all graphene membranes (Figure 4b–d) compared to the original graphene, as seen in Figure 2a. The kinetic energy was reduced for single layers with 4.3 and 7.1 pore size, while it increased for the multi-layer graphene. The non-bond energy required to break the molecules into component atoms slightly increased compared to the original, with a slight reduction in the case of multi-layer graphene compared to the single-layer membranes. The changes were not drastic, and the dynamics summary for the membranes showed similar behaviour under the pressure of 500 MPa, which proves the stability of the membrane for the water desalination process.

Pore density or porosity is one of the important factors in membrane function. The total pore density was calculated based on the Equation (1) [25].
(1)$$Porosity=\frac{Total\;pore\;volume}{Total\;volume}*100\%$$

For the membrane with an open-pore size of 4.3 Å, the porosity was equal to 11%, while for the membrane with a pore size of 7.1 Å, the porosity was equal 13.1%. The distance between pores horizontally was 9.840 Å and 9.942 Å vertically for both membranes.

Figure 5 shows the simulation box of the water desalination process, consisting of graphene membranes and H_2_O and NaCl molecules. Figure 5a shows the single-layer graphene with the original pores of 2.8 Å, which had the highest salt rejection among single-layer graphene under 500 MPa, and Figure 5b shows the three layers of graphene, which had the highest salt rejection (100%) under the minimum design pressure among all graphene membranes designed and studied in this work. The details are given in the results and discussion section.

Water and NaCl molecules were introduced to the graphene membrane to evaluate the performance of the designed membrane as a separator between H_2_O and NaCl molecules. A linear relationship between water wetting behavior and the microscopic interactions of hydrophilic surfaces allows water molecules to pass the membrane within 1 ns of simulation [26]. The three layers were organized for the multi-layer graphene membrane, from top to bottom, the first layer with a 7.1 Å pore size, second layer with a 4.3 Å pore size and the third layer with a 2.8 Å pore size, keeping the same porosity as calculated by Equation (1).

Figure 6 shows the interactions of water molecules with the graphene membrane at the beginning of the simulation study. Dougherty (1998) showed that the temperature and pressure affect the water hydrogen bond length, while the covalent bond lengths are more stable [27]. He discovered that the shorter hydrogen bond length gives stronger hydrogen bonding. Therefore, as the temperature increases, the hydrogen bond length increases, allowing the water molecules to pass through the graphene membrane. However, in this study, because of the high pressure applied on the H_2_O molecules in the membrane system, the hydrogen bonds become weak and long with smaller electron density, allowing the molecules to pass through the graphene membrane, as shown in Figure 6.

## 3. Results

### 3.1. Water Permeability

Water permeability was calculated using Equation (2):(2)$$Permeability=\frac{\left(membrane\;thickness\;\left(mm\right)\right)\left(amount\;of\;permeate\;\left(g\right)\right)}{\left(membrane\;surface\;area\;\left({cm}^{2}\right)\;\right)\left(time\;\left(s\right)\right)\left(differential\;pressure\;\left(bar\right)\right)}\;$$

Based on Equation (2) [28], the permeability was calculated for the duration of 1 ns to show the maximum numbers of H_2_O molecules that pass through the graphene membrane under different applied pressure values. For the single-layer graphene membrane, the membrane thickness was 3.4 Å, and the surface area of the membrane was 35.677 × 35.677 Å^2^. Each time, a different pore size was used under identical conditions. Figure 7a shows the water permeation rate through single and multi-layer graphene membranes versus different applied pressure values. As shown in Figure 7a, water permeability increases when the applied pressure increases for all designed membranes. There is a direct relationship between the water permeability and applied pressure and pore diameter. It reached 7.9 × 10^−9^ (mm.g.cm^−2^s^−1^.bar^−1^) under the applied pressure of 500 MPa for the single-layer membrane with a pore size of 7.1 Å. To better understand the nature of water flow, the percentages of water molecule permeation were calculated as a function of time using the method offered by other research studies [29]. For instance, at 0.5 ns, 55% of H_2_O molecules were filtered in the case of a single-layer graphene membrane with 7.1 Å pore under 500 MPa pressure, while for the graphene membrane with pore sizes of 4.3 and 2.8 Å, the percentage of H_2_O filtered molecules was 41% and 20%, respectively. In addition, at 0.8 ns, 73% of H_2_O molecules were filtered for the graphene membrane with 7.1 Å pore size, while it was 53% and 32% for 4.3 Å and 2.8 Å, respectively. For lower applied pressure values, the permeation rates were lower for all single and multi-layer membranes. The water molecules permeated at an approximately constant rate under the different applied pressure values during the simulation because of the low salt concentration, as also observed in ref. [30]. However, as the applied pressure increased, the salt rejection decreased, as shown in Figure 7b. Therefore, for the salt rejection, a specific combination of applied pressure and pore size should be selected to ensure the highest possible water permeability with an acceptable salt rejection rate.

The permeability was higher for the multi-layer G membrane than the single-layer graphene with the pore size of 2.8 Å (original graphene) under 500 MPa but lower than graphene with a pore size of 4.3 or 7.1 Å. This is due to surface hydrophilicity, which results in a slight increase in the permeability of H_2_O molecules through the graphene membrane [31,32]. At the beginning of the simulation (t = 0.2 ns), 15% of H_2_O molecules passed the first graphene layer with 7.1 Å, while at (t = 0.5 ns), 40% of H_2_O molecules passed the first G layer with 7.1 Å. In addition, for passing the second and third graphene layers, it took 0.1 ns for the first H_2_O molecule to pass through the second and third graphene layer with a pore size of 4.3 and 2.8 Å. However, at t = 0.5 ns, 25% of H_2_O molecules passed through three graphene layers, as the permeability rate increased until the end of the simulation study. However, at lower applied pressure values, approximately the same performance with a constant rate of H_2_O permeation through the multi-layer G membrane during the simulation study was observed.

### 3.2. Salt Rejection

Although the diameter of the nanopores of the membrane should be designed in such a way to allow water to pass through, the diameter must also be suitable for the salt rejection. Figure 7b shows the relationship between the salt rejection rate, applied pressure and pore size for the single and multi-layer membranes. The membrane’s susceptibility to salt rejection was verified by the effect of applied pressure on the ions through the membrane. The salt rejection rates were calculated using the salinity of the permeate solution at a certain time step during 1 ns relative to the initial salinity of the feed for the range of pore systems. Salt rejection rates calculated in terms of Na^+^ and Cl^−^ concentrations were based on Equation (3) [33].
(3)$$R=\frac{Nf-Np}{Nf}$$
where *R*, *N_p_* and *N_f_*, are the salt rejection rate, ion concentration in the permeate and feed sides, respectively.

For the single-layer graphene membrane, the results showed a higher salt rejection in lower applied pressures and vice versa. This may be due to path change of the high content of salts within the permeate that decreases the salt rejection efficiency [34]. These results can be used to compare the effect of different pore sizes and pressures on simulated membrane performance. Among the single-layer G membranes, the highest salt rejection (92%) was achieved under the applied pressure of 100 MPa for graphene with a pore size of 2.8 Å without any functionalization.

However, for a multi-layer G membrane, almost 100% salt rejection is achieved under all types of applied pressure (Figure 7b). For single-layer graphene, as the pressure increased, the salt rejection reduced. For instance, under 500 MPa, only 60% of NaCl molecules were blocked by the first layer with a pore size of 7.1 Å, while the second layer blocked more than 90% of NaCl molecules, and the third layer blocked the rest during 1 ns of simulation.

Comparing the designed membranes in this study with other reported articles was not easy, as each study used different material and under various conditions. However, the results of salt rejection and water permeability in this study were compared with the similar studies available in literature, as summarized in Table 2.

To understand the behavior of the membrane under lower pressure, the performance of single-layer graphene with 2.8 Å pores and multi-layer graphene under 50 MPa were studied. This was to show that the simulation study can be applied under realistic conditions. As can be seen, compared to the three-layer graphene designed by the researcher [36] with the 98.35% salt rejection, our study shows 100% rejection with three graphene layers using a different design.

### 3.3. Water Permeability Profile after Several Cycles

Flux or water flux is generally defined as volume per area per unit of time. Water flux is used to show the water permeation rate through a desalination membrane. Gallons per square foot per day (GSFD) or liters per square meter per hour (L/m^2^/h) are units of flux. Figure 8 shows the water flux rates after a specific number of iterations for five consecutive uses of the same membrane for water desalination; it can be observed that after the fourth cycle, the flux became stable. The water flux in this study was calculated after a different number of iterations, which also shows the relation between the water flux and its stability after each cycle. The durability of the graphene membrane was checked under 500 MPa pressure by repeating the simulation method for five cycles under the same conditions during 5 ns as total simulation time. After 3 ns, the water flux became stable, while the salt rejection percentages did not change during all the cycles. Under the identical conditions, it could be observed that the water flux increased after each cycle until the fourth cycle and remained the same after the fourth cycle for a specific time.

## 4. Conclusions

The single and multi-layer graphene membranes were designed and simulated for water desalination through molecular dynamics (MD) using ReaxFF module of Amsterdam Modeling Suite (AMS) software. The effect of the pore size and applied pressure on the water permeability and salt rejection rate was studied to show the optimum conditions of the water desalination process through the designed graphene membrane. The single-layer graphene membrane showed salt rejection of 95% and 82% having a water permeability 0.347 × 10^−9^ and 2.94 × 10^−9^ (mm.g.cm^−2^s^−1^.bar^−1^), under the applied pressure of 50 and 500 MPa, respectively. As the applied pressure increases, the salt rejection decreases; therefore, a specific combination of applied pressure and pore diameter should be selected in order to have the highest possible water permeability with an acceptable salt rejection rate. In the multi-layer graphene membrane, the salt rejection reached 100%, with the water permeability equal to 3.4 × 10^−9^ (mm.g.cm^−2^s^−1^.bar^−1^). In addition, the performance of the graphene membrane improved even after reusing, which showed a higher water flux with the same rate of salt rejection. It could be observed that the flux rate increased after each cycle until the fourth cycle and after that remained stable. The results are promising and could be used to design a graphene membrane that is stable under high pressure up to 500 MPa.

## Figures and Tables

**Figure 1 membranes-12-01038-f001:**
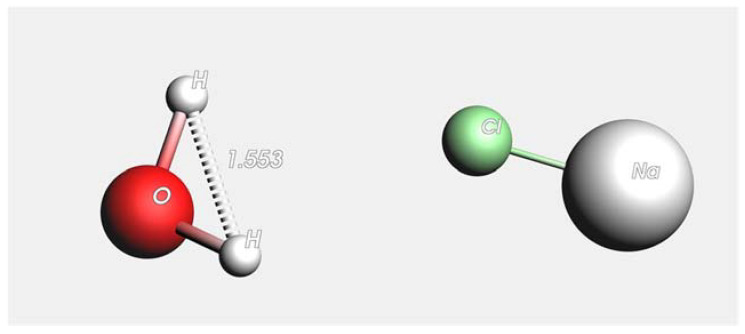
Dimensions of H_2_O and NaCl molecules (Å). Water molecules (H_2_O atom in red and white and NaCl atom in white and green).

**Figure 2 membranes-12-01038-f002:**
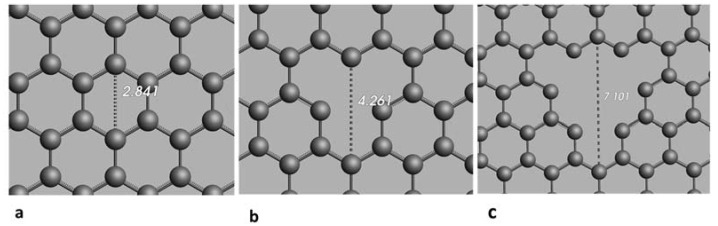
Pore size of nanopores created in G membrane in Å (**a**) 2.841 Å, (**b**) 4.261 Å and (**c**) 7.101 Å.

**Figure 3 membranes-12-01038-f003:**
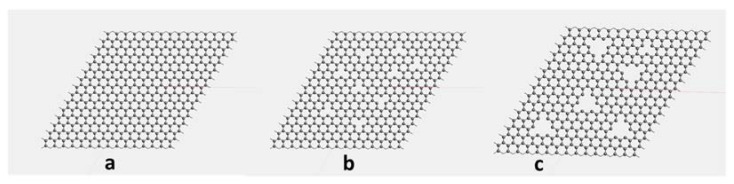
The structure of graphene membranes with different pore sizes of (**a**) 2.841 Å, (**b**) 4.261 Å and (**c**) 7.101 Å.

**Figure 4 membranes-12-01038-f004:**
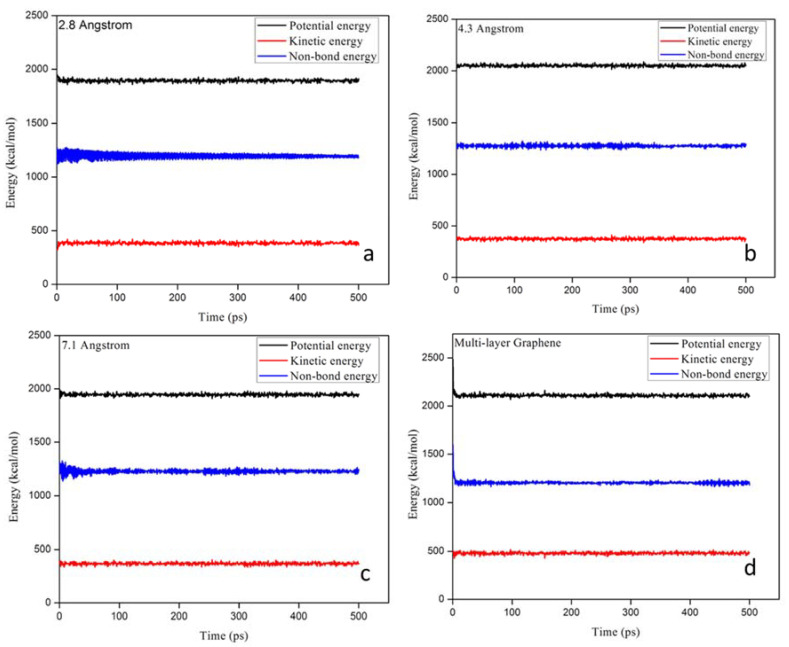
The dynamic summary for the simulated G membranes with different pore sizes during 0.5 ns simulation time, (**a**) membrane with original pore size of 2.8 Å, (**b**) membrane with 4.3 Å pore, (**c**) membrane with 7.1 Å pore and (**d**) mullti-layers membrane.

**Figure 5 membranes-12-01038-f005:**
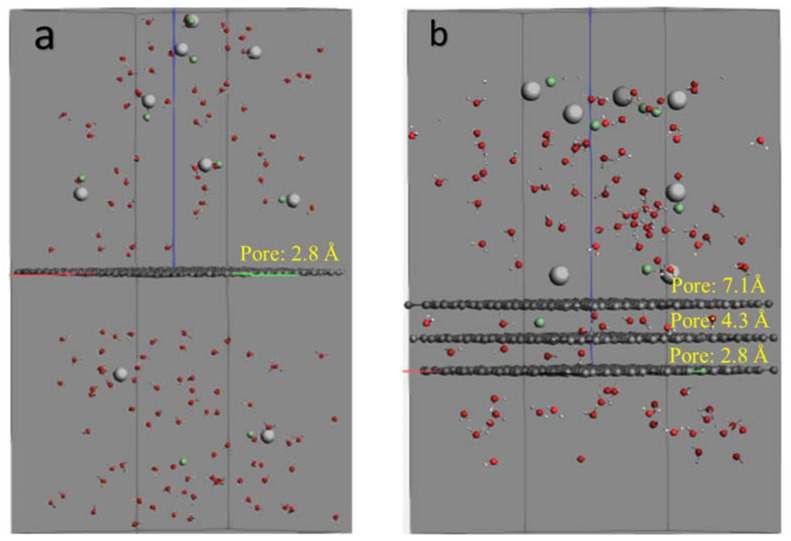
Simulation box consisting of graphene membrane with carbon atoms in grey, water molecules (H_2_O atoms) in red and white, and NaCl molecules in white and green. (**a**) Single-layer graphene membrane and (**b**) multi-layer graphene membrane at the beginning of simulation.

**Figure 6 membranes-12-01038-f006:**
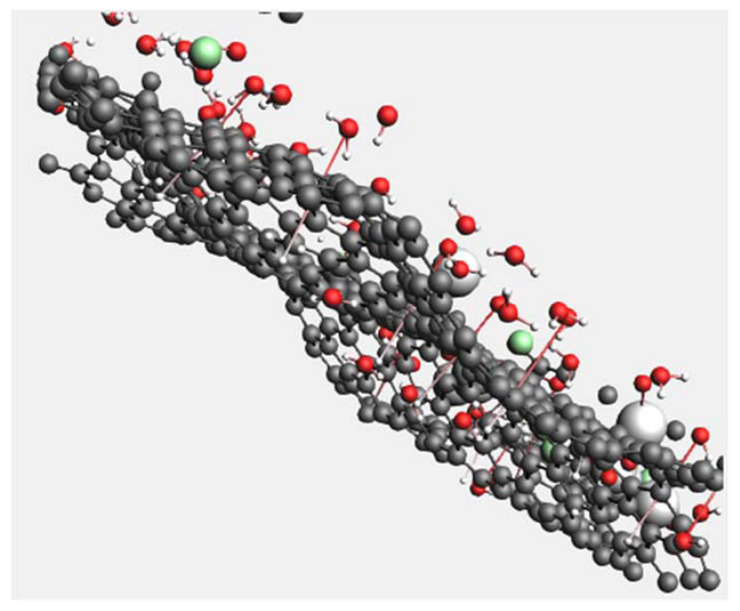
Snapshot of the graphene membrane with water molecules at the beginning of simulation: graphene membrane (carbon atoms in grey), water molecules (H_2_O atoms in red and white), and NaCl molecules (in white and green).

**Figure 7 membranes-12-01038-f007:**
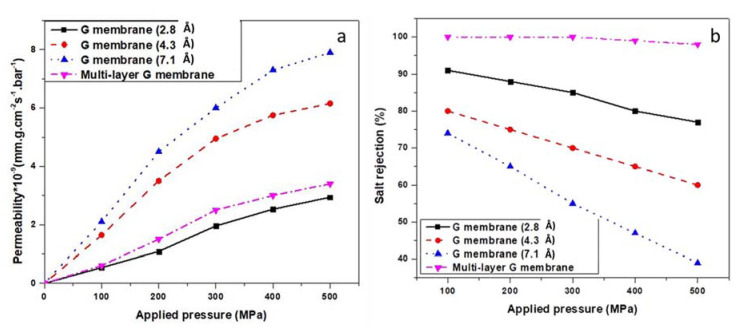
(**a**) Average water permeability through simulated G membrane; (**b**) Salt rejection percentages under different applied pressure values for single and multi-layer membranes at t = 1 ns.

**Figure 8 membranes-12-01038-f008:**
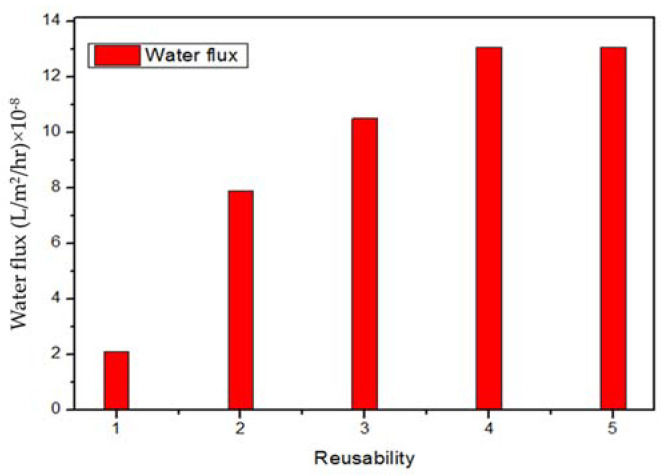
Relationship between the rates of water flux in the designed membrane with a specific number of iterations for five cycles (each cycle duration is 1 ns under 500 MPa pressure).

**Table 1 membranes-12-01038-t001:** The recorded energy level for the G membrane before and after applying 500 MPa pressure.

Membrane Layer	Pore Size (Å)	Potential Energy (kcal/mol)		Kinetic Energy (kcal/mol)		Non-Bond Energy (kcal/mol)	
		Initial	Under Pressure	Initial	Under Pressure	Initial	Under Pressure
Single layer	2.8	1698	1912	384	365	1270	1190
Single layer	4.3	2039	2049	374	375	1500	1290
Single layer	7.1	2038	1945	368	369	1310	1240
Multi-layer	7.1, 4.3, 2.8	2578	2102	480	487	1588	1221

**Table 2 membranes-12-01038-t002:** Comparison of the results of this study with the available literature.

Simulation Method	Applied Pressure (MPa)	Pore Size (Å)	Salt Rejection (%)	Water Permeability	Material	References
NPT Berendsen	50	2.8	95 (285 NaCl molecules blocked)	0.347 × 10^−9^ (mm.g.cm^−2^s^−1^.bar^−1^) 106 H_2_O molecules filtered	Single layer graphene	This paper
NPT Berendsen	500	2.8	77 (231 NaCl molecules blocked)	2.94 × 10^−9^ (mm.g.cm^−2^s^−1^.bar^−1^) 900 H_2_O molecules filtered	Single layer graphene	This paper
NPT Berendsen	500	4.3	60 (180 NaCl molecules blocked)	6.154 × 10^−9^ (mm.g.cm^−2^s^−1^.bar^−1^) 2430 H_2_O molecules filtered	Single layer graphene	This paper
NPT Berendsen	500	7.1	39 (117 NaCl molecules blocked)	7.9 × 10^−9^ (mm.g.cm^−2^s^−1^.bar^−1^) 2850 H_2_O molecules filtered	Single layer graphene	This paper
NPT Berendsen	100	2.8	91 (273 NaCl molecules blocked)	0.541 × 10^−9^ (mm.g.cm^−2^s^−1^.bar^−1^) 180 H_2_O molecules filtered	Single layer graphene	This paper
NPT Berendsen	100	7.1, 4.3, 2.8	100 (All NaCl molecules blocked)	0.61 × 10^−9^ (mm.g.cm^−2^s^−1^.bar^−1^) 197 H_2_O molecules filtered	Multi-layer graphene membrane	This paper
NVE integration	1.1 × 10^−4^	3	Not reported	Only 1 H_2_O molecules passed	Graphene	[35]
NVE integration	240	6	Not reported	24 H_2_O molecules passed	Graphene	[35]
NVE integration	140	8	Not reported	56 H_2_O molecules passed	Graphene	[35]
NVT ensemble	100	Not reported	5 ions permeated	900–1000 (water molecules filtered)	Graphene	[36]
NVT ensemble	Rigid piston	3.3	98.35	31.9 (g/s cm^2^)	Three graphene layer membrane	[19]
NVT ensemble	Rigid piston	4	68.4	50.5 (g/s·cm^2^)	Three graphene layer membrane	[19]
NVT ensemble	Rigid piston	5	55	55.7 (g/s·cm^2^)	Three graphene layer membrane	[19]

## Data Availability

Data are available upon request from the corresponding author and based on university rules.

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
