# Peer review of "Design of Multi-Layer Graphene Membrane with Descending Pore Size for 100% Water Desalination by Simulation Using ReaxFF"

_membranes, 2022, doi:10.3390/membranes12111038_

Round 1
Reviewer 1 Report
The manuscript entitled “Design of multi-layer graphene membrane with descending 2 pore size for 100% water desalination by simulation using ReaxFF” reports an interesting study of the desalination process of water using graphene. The methodology is based on determining the water permeability under different nanopores and pressures.
From my point of view, the manuscript deserves to be published on membranes. However, there are a few issues to be solved.
1. Several mistakes in the writing process have to be carefully corrected. For example, lines 98-112 and 232-235 have to be deleted.
2. In line 145, it is mentioned that the total length of the molecules is 10.35 A. Can you explain this quantity or correct it?
3. It is used non-bonded energy. Please, be more specific about it and its physical significance.
4. I recommend including some highlights of the NPT Berendsen. Why is this method best than others?
5. The authors assert that “the hydrogen bonds become weak and long, allowing the molecules to pass through the graphene membrane” this is a good result, and it is essential to clarify this point. How do you measure the weakness of the bond? What is the distortion of the bond length?
6. About the multi-layer configuration, it is unclear whether the pores are aligned or centered. Please clarify this point.
Author Response
Author's Reply to the Review Report (Reviewer 1)
The manuscript entitled “Design of multi-layer graphene membrane with descending 2 pore size for 100% water desalination by simulation using ReaxFF” reports an interesting study of the desalination process of water using graphene. The methodology is based on determining the water permeability under different nanopores and pressures.
From my point of view, the manuscript deserves to be published on membranes. However, there are a few issues to be solved.
Thank you for the comments, we have edited the paper based on your comments and improve the quality of the paper.
- Several mistakes in the writing process have to be carefully corrected. For example, lines 98-112 and 232-235 have to be deleted.
The additional writing which was there by mistake deleted.
- In line 145, it is mentioned that the total length of the molecules is 10.35 A. Can you explain this quantity or correct it?
The quantity has been deleted. The text has been corrected
- It is used non-bonded energy. Please, be more specific about it and its physical significance.
More discussion has been added about the non-bonded energy.
- I recommend including some highlights of the NPT Berendsen. Why is this method best than others?
More discussion has been added about the specifications of the NPT Berendsen method.
- The authors assert that “the hydrogen bonds become weak and long, allowing the molecules to pass through the graphene membrane” this is a good result, and it is essential to clarify this point. How do you measure the weakness of the bond? What is the distortion of the bond length?
The weakness of the bond has been measured by calculating the electron density of the bond which was large (Strong bond) and then became small (Weak bond). The statement has been added to the paragraph.
- About the multi-layer configuration, it is unclear whether the pores are aligned or centered. Please clarify this point.
The pores created randomly and are not aligned.
Author Response
The review paper entitled “Design of multi-layer graphene membrane with descending pore size for 100% water desalination by simulation using Re- axFF” by Rokhsareh and the coauthor, comprehensively studied multi-layer graphene membrane with descending pore size.
I think it is of great interest in the community of membrane science, because water desalination is the key application of Membrane separation (MS) technology. As a result, I will recommend the publication of this manuscript in the current form.
Thank you for your kind comment. We have corrected the manuscript based on the comments and the journal standards.
Reviewer 3 Report
Reviewer report on manuscript Membranes-1882015
Ibrahima et al. “Design of multi-layer graphene membrane with descending pore size for 100% water desalination by simulation using ReaxFF”
In this manuscript, a molecular dynamics (MD) simulation was performed using ReaxFF software to simulate water desalination efficiency by single and multi-layer graphene membrane. The graphene membrane with different pore sizes and a multi-layer graphene membrane with descending pore size in each layer were designed and studied under different pressures.
The manuscript can be accepted after minor revision. Authors should make the following corrections:
1. Section 1 “Introduction” should be corrected. The text between lines 97 and 112 should be removed. There are rules for a manuscript preparation.
2. Section 3 “Results” should be corrected. The text between lines 233 and 235 should be removed. There are rules for a manuscript preparation.
3. List of references must be formatted in the same style for all references.
4. The references list should be extended and bring up-to date (2022) in this field, since 100% of references are older than 2019.
5. The choice of models for multilayer graphene membranes’ simulation should be better grounded with references to the up-to date researches in the field of multilayer graphene synthesis, including ACS Nano, 2021, 15, 12358 and Carbon 2022, 194, 52.
6. Numerous typos and missing hyphens, periods and spaces must be corrected, e.g.:
Page 3, Line 140 (Materials and methods)
Page 3, Line 141 (Materials and methods)
Page 4, Line 159 (Materials and methods)
Page 5, Line 178 (Materials and methods)
Page 6, Line 189 (Materials and methods)
Page 6, Line 194 (Materials and methods)
Page 7, Line 245 (Results)
Page 8, Line 269 (Results)

Author Response
Ibrahima et al. “Design of multi-layer graphene membrane with descending pore size for 100% water desalination by simulation using ReaxFF”
In this manuscript, a molecular dynamics (MD) simulation was performed using ReaxFF software to simulate water desalination efficiency by single and multi-layer graphene membrane. The graphene membrane with different pore sizes and a multi-layer graphene membrane with descending pore size in each layer were designed and studied under different pressures.
The manuscript can be accepted after minor revision. Authors should make the following corrections:
Thank you for your kind comments, the manuscript has been reviewed extensively based on the comments and has been improved further.
- Section 1 “Introduction” should be corrected. The text between lines 97 and 112 should be removed. There are rules for a manuscript preparation.
Apologize for the mistake. Those extra text have been deleted.
- Section 3 “Results” should be corrected. The text between lines 233 and 235 should be removed. There are rules for a manuscript preparation.
The extra text has been deleted.
- List of references must be formatted in the same style for all references.
The references have been formatted in the same style.
- The references list should be extended and bring up-to date (2022) in this field, since 100% of references are older than 2019.
The new references have been added to the paper
- The choice of models for multilayer graphene membranes’ simulation should be better grounded with references to the up-to date researches in the field of multilayer graphene synthesis, including ACS Nano, 2021, 15, 12358 and Carbon 2022, 194, 52. The suggested papers have been studied but those have the different perspective.
- Numerous typos and missing hyphens, periods and spaces must be corrected, e.g.:
Page 3, Line 140 (Materials and methods) Deleted
Page 3, Line 141 (Materials and methods) Deleted
Page 4, Line 159 (Materials and methods) The text has been corrected
Page 5, Line 178 (Materials and methods) The text has been edited
Page 6, Line 189 (Materials and methods) The text has been edited
Page 6, Line 194 (Materials and methods) The text has been edited
Page 7, Line 245 (Results) Checked
Page 8, Line 269 (Results) The text has been edited
Round 2
Reviewer 1 Report
There is a significant improvement in the presentation of the results and methodology of the manuscript. In my opinion, it can be published in its present form.